# Adenosine Increases the Immunosuppressive Capacity of Cervical Cancer Cells by Increasing PD-L1 Expression and TGF-β Production through Its Interaction with A_2A_R/A_2B_R

**DOI:** 10.3390/ph17030397

**Published:** 2024-03-19

**Authors:** Rosario García-Rocha, Alberto Monroy-García, Ana Luisa Vázquez-Cruz, Luis Antonio Marín-Aquino, Benny Weiss-Steider, Jorge Hernández-Montes, Christian Azucena Don-López, Gabriela Molina-Castillo, María de Lourdes Mora-García

**Affiliations:** 1Laboratorio de Inmunobiología, Unidad de Investigación en Diferenciación Celular y Cáncer-UMIEZ, FES-Zaragoza, Universidad Nacional Autónoma de México, Ciudad de México 04510, Mexico; rgrbiologia@hotmail.com (R.G.-R.); albertomong13@comunidad.unam.mx (A.M.-G.); anavazc00@gmail.com (A.L.V.-C.); bennyweiss@hotmail.com (B.W.-S.); jhzms@yahoo.com (J.H.-M.); damostwanted060993@live.com.mx (C.A.D.-L.); gmolina2509@comunidad.unam.mx (G.M.-C.); 2Laboratorio de Inmunología y Cáncer, Unidad de Investigación Médica en Enfermedades Oncológicas, Hospital de Oncología, CMN SXXI, Instituto Mexicano del Seguro Social, Ciudad de México 06720, Mexico; bioluma89@gmail.com; 3Doctorado en Ciencias en Inmunología, Instituto Politécnico Nacional, Ciudad de México 07738, Mexico; 4Programa de Posgrado en Ciencias Biológicas, Universidad Nacional Autónoma de México, Ciudad de México 04510, Mexico

**Keywords:** adenosine, A_2A_R and A_2B_R antagonists, cervical cancer, immunosuppression, PD-L1 expression, TGF-β1, CD8+ T lymphocytes

## Abstract

The present study provides evidence showing that adenosine (Ado) increases the expression of programmed death ligand 1 (PD-L1) in cervical cancer (CeCa) cells by interacting with A_2A_R/A_2B_R and that TGF-β1 acts in an autocrine manner to induce PD-L1 expression, enhancing the immunosuppressive effects of CeCa cells on activated T lymphocytes (ATLs) and CD8+ cytotoxic T lymphocytes (CTLs) specific for antigenic peptides derived from E6 and E7 proteins of HPV-16. Interestingly, the addition of the antagonists ZM241385 and MRS1754, which are specific for A_2A_R and A_2B_R, respectively, or SB-505124, which is a selective TGF-β1 receptor inhibitor, to CeCa cell cultures significantly inhibited PD-L1 expression. In addition, supernatants from CeCa cells that were treated with Ado (CeCa-Ado Sup) increased the expression of PD-1, TGF-β1, and IL-10 and decreased the expression of IFN-γ in ATLs. Interestingly, the addition of an anti-TGF-β neutralizing antibody strongly reversed the effect of CeCa-Ado Sup on PD-1 expression in ATLs. These results strongly suggest the presence of a feedback mechanism that involves the adenosinergic pathway, the production of TGF-β1, and the upregulation of PD-L1 expression in CeCa cells that suppresses the antitumor response of CTLs. The findings of this study suggest that this pathway may be clinically important and may be a therapeutic target.

## 1. Introduction

Cervical cancer (CeCa) ranks fourth in incidence and mortality among all types of cancer worldwide. In 2020, approximately 600,000 new cases of CeCa and more than 340,000 deaths due to CeCa were recorded. According to data from the WHO, approximately 90% of deaths associated with CeCa occur in developing countries [1,2]. Persistent infection caused by high-risk human papillomaviruses (HR-HPV), such as the HPV-16 and HPV-18 genotypes, which are present in approximately 50% and 20% of CeCa patients, respectively, can trigger progression from low-grade cervical intraepithelial neoplasia (CIN I) to high-grade neoplasia intraepithelial cervical (CIN II-III) and eventually to CeCa [3]. The immune response against HPV antigens can eliminate most precursor infections and lesions; however, some women infected with HR-HPV genotypes develop cancer, suggesting that other risk factors may be involved [4].

Several studies have proposed that immunosuppression mechanisms may play a relevant role in the development of CeCa. In this context, several findings have shown that programmed death ligand 1 (PD-L1), which is expressed on the surface of different types of cancer cells, can induce immunosuppressive mechanisms to enable tumors to evade the host’s anticancer immune responses [5,6]. Additionally, studies have shown that the immune checkpoint receptor of PD-L1, namely, programmed cell death 1 (PD-1), which is primarily expressed on T cells, monocytes, macrophages, and natural killer cells, could be a critical negative regulator of cancer biology and may have the ability to support cancer development, growth, invasion, and metastasis [7]. PD-L1 expression on the surface of tumor cells can attenuate the ability of the immune system to successfully eliminate tumor cells because the binding of the PD-1 receptor to its ligands, namely, PD-L1 and/or PD-L2, negatively regulates T-cell receptor (TCR) signaling, resulting in the inhibition of cytotoxic T lymphocytes [8]. In this sense, blocking the PD-L1/PD-1 pathway through the use of therapeutic antibodies has shown important antitumor effects in patients with advanced cancers [9]. However, the response rates to anti-PD-L1 therapy in several solid tumors have been limited, with a response ranging between 10% and 40% [10]. Therefore, understanding the regulation of PD-L1 expression is relevant for improving the efficacy of immune checkpoint blockade-based immunotherapy. Recent studies have reported that the expression of PD-L1 is intricately regulated by various processes, such as gene transcription, posttranscriptional and posttranslational modifications (PTMs), and exosomal transport [11]. Among the mechanisms that regulate PD-L1 expression, the production of cytokines, such as IL-1a, IL-27, IL-10, and IL-32g, in the tumor microenvironment (TME) can upregulate the expression of PD-L1 on monocytes and tumor cells [12,13]. Similarly, some studies have shown that TGF-β also induces the expression of PD-L1 in different tumors [14,15,16], as well as in human and murine fibroblasts in a Smad2/3- and YAP/TAZ-dependent manner [17]. In addition, TGF-β plays a crucial role in the development of CeCa by establishing a local immunosuppressive microenvironment in the cervix during HR-HPV infection and inhibiting the proliferation and activation of T lymphocytes with antitumor activity [18]. In the particular case of CeCa, a tendency to increase the expression of PD-L1 has been observed in CIN or CeCa patients with HR-HPV infection [19,20,21], a previous history of chemotherapy [20], metastatic tumors [21], a history of multiple parity and abortion [22], an advanced stage [23], poorly differentiated tumors [24], and large tumors [24]. In addition, studies have revealed that the expression of PD-L1 and PD-1 in cervical DCs and T cells, respectively, is associated with HR-HPV positivity and with the degree of CeCa development [25].

Moreover, recent reports indicate that the adenosinergic pathway plays an important role in the pathogenesis of gynecological cancers, including CeCa [26,27]. Through this process, ATP/ADP nucleotides, which are present at high concentrations (>100 μM) in the TME [28], are converted to adenosine monophosphate (AMP) via the hydrolytic activity of the ectoenzyme CD39 (ectonucleoside triphosphate diphosphohydrolase-1, ENTPD1; EC 3.6.1.5) and later to Ado by 5′-ectonucleotidase (CD73, EC 3.1.3.5) [29,30]. The main signaling effects of Ado are mediated by specific adenosine receptors (ARs) that are localized in the membranes of target cells, where they are coupled to G proteins. These ARs are divided into four subtypes: A_1_R, A_2A_R, A_2B_R, and A_3_R [31]. In the TME, Ado exerts immunosuppressive effects on CTLs, dendritic cells, NK cells, and B cells through interactions with the high-affinity receptor A_2A_R [32]. In addition, through interaction with A_1_R, A_2A_R, and A_2B_R, Ado can also promote tumor growth by inducing the proliferation, invasion, and metastasis of tumor cells [33,34]. For these reasons, adenosine signaling has emerged as a target in cancer immunotherapy due to its key role in the metabolic pathway that regulates tumor immunity [34]. In previous studies, we reported that Ado induces the production of TGF-β1 by CeCa cells by interacting with ARs, specifically A_2A_R and A_2B_R [35], and that this cytokine strongly contributes to enhancing protumoral characteristics related to tumor progression, such as migration and invasion, immune evasion, immunosuppression, and chemoresistance [36]. Considering these findings, in this study, we analyzed the role of Ado in the induction of PD-L1 expression in CeCa cells as well as the ability of these cells to inhibit the activation and cytotoxic activity of CD8+ T lymphocytes. We found that Ado induced PD-L1 expression in CeCa cells through its interaction with A_2A_R/A_2B_R and the production of TGF-β. In addition, supernatants derived from CeCa cells that were treated with Ado increased the expression of PD-1, TGF-β1, and IL-10 and decreased the expression of IFN-γ in activated CD8+ T cells, suggesting that this pathway may be clinically important and may be a therapeutic target in CeCa.

## 2. Results

### 2.1. Ado Induces PD-L1 Expression in CeCa Cells through Its Interaction with A_2A_R/A_2B_R and the Autocrine Production of TGF-β1

Our research group previously reported that CeCa cells inhibit the effector functions of CD8+ cytotoxic T lymphocytes through the production of high amounts of Ado [37] and that this nucleoside induces the expression and production of TGF-β1 in CeCa cells by interacting with A_2A_R/A_2B_R [35]. Because TGF-β1 is associated with the expression of PD-L1 in different tumors [14,15,16], in this study, we analyzed the effect of Ado on the production of TGF-β1 and on the induction of PD-L1 expression in CeCa cells. To this end, 1 × 10^5^ CaSki and HeLa cells were cultured in the presence of 0.1 mM or 1 mM Ado for 72 h. The baseline expression of PD-L1 in CeCa cells served as the Ctl (Figure 1a,b). Ado increased the expression of PD-L1 in both cell lines, and this effect depended on the duration of culture with Ado and the concentration of this nucleoside. The highest expression of this molecule was detected after culture with 1 mM Ado for 72 h (Figure 1a,b). Therefore, these cell culture conditions were used in subsequent experiments.

To analyze whether A_2A_R or A_2B_R participates in the Ado signaling-induced expression of PD-L1, CeCa cells were cultured for 72 h in the presence or absence of 10 µM ZM241385 and MRS1754, which are selective antagonists for A_2A_R and A_2B_R, respectively, and in the presence or absence of 1 mM Ado. Blocking A_2A_R significantly reduced the expression of PD-L1 in CaSki and HeLa cells; however, blocking A_2B_R significantly reduced the expression of PD-L1 in only HeLa cells (Figure 2a and Appendix A). To confirm the participation of A_2A_R and A_2B_R in the signaling-induced expression of PD-L1, CeCa cells were cultured for 72 h in the presence of 10 μM CGS21680, a selective A2 adenosine receptor agonist. Interestingly, CGS21680 induced a significant increase in PD-L1 expression in CaSki and HeLa cells, and this increase was comparable to that observed in CeCa cells cultured in the presence of 1 mM Ado (Appendix A).

In contrast, CaSki and HeLa cells cultured in the presence of 1 mM Ado exhibited increased TGF-β1 production. In fact, after 24 h of culture, the TGF-β1 concentrations in the supernatants of both CeCa cell lines were significantly greater than those in the supernatants of cells that were cultured without Ado (black bars), which were used as a Ctl. However, although the levels of TGF-β1 in the CaSKi cell supernatants remained high during 72 h of culture, the levels of this cytokine in the HeLa cell supernatants decreased significantly after 24 h of culture (Figure 2b). 

To determine whether TGF-β1 induces the expression of PD-L1, CeCa cells were cultured for 24, 48, and 72 h in the presence of 20 ng/mL recombinant human TGF-β1 (rHuTGF-β1). Initially, the expression of the TGF-β1 type 2 receptor (TGF-β1RII)) in CeCa cells was analyzed. The baseline expression of TGF-β1RII in HeLa cells was 4 times greater than that in CaSki cells, and the mean fluorescence intensity (MFI) values obtained for these cells were 3631 and 892, respectively. However, the culture of CaSki and HeLa cells for 72 h in the presence of 1 mM Ado doubled the expression of TGF-β1RII, as revealed by MFI values of 6313 and 2130, respectively (Appendix A). After 24 h of culture, the expression of PD-L1 was significantly increased in both cell lines; however, over time, the expression of PD-L1 gradually decreased (Figure 2c). To determine whether TGF-β1 acts in an autocrine manner to induce PD-L1 expression, CeCa cells were cultured in the presence or absence of 1 mM Ado and in the presence or absence of 25 μM SB-505124, which is a selective inhibitor of TGF-β1R. The addition of SB-505124 to the cell cultures reversed the Ado-induced expression of PD-L1 by more than 80% in CaSki cells and by approximately 40% in HeLa cells (Figure 2d). These results suggest that the production of TGF-β1 after the interaction of Ado with A_2A_R and A_2B_R is important for inducing PD-L1 expression in CeCa cells.

### 2.2. Ado-Treated CeCa Cells Inhibit the Proliferation of CD8+ T Lymphocytes through PD-L1 Expression

To analyze the PD-L1-mediated immunosuppressive effect of CeCa cells, CaSki and HeLa cells were cultured in the presence or absence of 1 mM Ado (CaSki-Ado and HeLa-Ado) for 72 h. The cells were subsequently fixed with paraformaldehyde, washed, and cocultured with ATLs at a CeCa cell:ATL ratio of 1:5 for 96 h. The level of proliferation exhibited by ATLs that were cultured alone was considered the positive control level. The proliferation indices (PIs) of the ATLs that were determined at the beginning of the culture were set to 1. After 96 h of culture, the PIs of the ATLs were 2.25–2.5 (Figure 3a,b, black line). The addition of CaSki or HeLa cells to the cultures blocked the proliferation of ATLs; in fact, after 96 h of cocultivation, the PIs of the ATLs were 1.75 and 1, respectively (Figure 3a,b, red line). The addition of CaSki-Ado or HeLa-Ado to ATL cultures resulted in greater inhibition of ATL proliferation; after 96 h of coculture, the PIs were 0.9 and 0.7, respectively (Figure 3a,b, blue line). Interestingly, the addition of anti-PD-L1 antibodies to the ATL cultures significantly reversed the CaSki-Ado- and HeLa-Ado-mediated inhibition of ATL proliferation, with PIs of 2.25 and 1.5, respectively (Figure 3a,b, green line). Importantly, CeCa cells that were cultured in the presence or absence of Ado and then cultured with ATLs did not proliferate when they were cultured alone for 96 h. These results suggest that Ado strongly increases the immunosuppressive capacity of CeCa cells through the induction of PD-L1 expression. To corroborate this effect, CaSki cells or CaSki-Ado cells, previously fixed with paraformaldehyde, were cocultured for 48 h at a ratio of 1:5 with CTLs that were previously stimulated with antigenic peptides derived from the E6 and E7 proteins of HPV-16, which specifically bind to the HLA-A2 allele and which we previously analyzed in CaSki cells [38,39,40]. A neutralizing anti-PD-L1 antibody was added to some cocultures. CTLs cultured in the absence of CaSki cells were used as a control (Ctl). Subsequently, CaSki cells labeled with CFSE and 7-AAD were challenged with effector cells at a ratio of 1:2. Interestingly, the coculture with CaSki or CaSki-Ado reduced the cytotoxic activity of CTLs on CaSki cells by approximately 50% compared with the Ctl; however, the addition of anti-PD-L1 antibodies to the cocultures blocked the suppressive effect of CaSki or CaSki-Ado cells (Figure 4).

### 2.3. Supernatants Derived from CeCa/Ado Cells Strongly Suppress the Activation of CD8+ T Lymphocytes

Increased expression of PD-1/PD-L1 pathway components is correlated with impaired cell-mediated immunity in patients with HR-HPV/related cervical intraepithelial neoplasia [25]. In fact, the proportion of Th1/Th2 cytokines in cervical exudates is correlated with HR-HPV positivity and the CIN grade [41]. The induction of PD-1/PD-L1 can also be regulated by TGF-β [25,42], and Ado induces TGF-β production by CeCa cells [35]. In this study, we also analyzed the effect of supernatants (Sup) from CeCa and CeCa-Ado cells on CD8+ T lymphocyte activation. For this purpose, 2.5 × 10^5^ CD8+ T cells were cultured for 48 h with beads coated with anti-CD2/CD3/CD28 antibodies at a ratio of 2:1 with or without 30% Sup from CeCa cells or CeCa-Ado cells and with or without 1 μg/mL neutralizing anti-TGF-β1 antibody. The expression level of PD-1 in ATLs that were cultured without CeCa or CeCa-Ado Sup was considered the Ctl level. Compared with the Ctl treatment, treatment with Sup derived from CeCa cells increased PD-1 expression in ATLs by 30–60%, whereas treatment with Sup derived from CeCa-Ado cells increased PD-1 expression by approximately 100%. Interestingly, the addition of an anti-TGF-β neutralizing antibody strongly reversed (more than 70%) the CeCa-Ado Sup-induced increase in PD-1 expression by ATLs (Figure 5a).

Additionally, the effects of Sup derived from CeCa or CeCa-Ado cells on IFN-γ, TGF-β, and IL-10 production by ATLs were analyzed. Interestingly, Sup from CeCa and CeCa-Ado cells strongly inhibited the production of IFN-γ by ATLs compared with the IFN-γ production by ATLs that were cultured without Sup from CeCa or CeCa-Ado cells (Ctl). However, the addition of an anti-TGF-β neutralizing antibody did not reverse the inhibitory effect of Sup on IFN-γ production by ATLs (Figure 5b). In contrast, Sup from CeCa cells increased the expression of TGF-β (Figure 5c) and IL-10 (Figure 5d) in ATLs by 20–30%, whereas Sup from CeCa-Ado cells increased the expression of TGF-β (Figure 5c) and IL-10 (Figure 5d) in ATLs by 40–80%. The addition of an anti-TGF-β neutralizing antibody reversed the CeCa-Ado cell-derived Sup-induced increase in TGF-β and IL-10 expression by more than 50% (Figure 5c,d). To confirm the effects of TGF-β on the expression of PD-1, IFN-γ, IL-10, and TGF-β in ATLs, CD8+ T lymphocytes were cultured for 48 h in the presence of beads coated with anti-CD2/CD3/CD28 antibodies at a ratio of 2:1 and in the presence or absence of 20 ng/mL rHuTGF-β. We observed an increase in the expression of PD-1 of more than 60% (Figure 6a), a reduction in the expression of IFN-γ of more than 20% (Figure 6b), and increases in the expression of both TGF-β by more than 50% (Figure 6c) and IL-10 by more than 40% (Figure 6d) in ATLs that were cultured in the presence of rHuTGF-β1 compared with the levels in ATLs that were cultured without this cytokine. Our results suggest that adenosine increases the immunosuppressive capacity of CeCa cells via the production of TGF-β, which acts in an autocrine manner to induce the expression of PD-L1 on CeCa cells and acts in a paracrine manner to inhibit the activation of CD8+ lymphocytes through the induction of PD-1 expression, resulting in decreased production of IFN-γ and increased production of TGF-β and IL-10.

## 3. Discussion

Several studies have shown that the expression of PD-L1/PD-1 is associated with HPV-related cancer, especially cancers associated with HPV-16 and HPV-18, which account for approximately 70% of cervical cancer cases [19,20,21]. It has been reported that the HPV E5/E6/E7 oncogenes activate multiple signaling pathways, including the hypoxia-inducible factor 1α, STAT3/NF-kB, PI3K/AKT, MAPK, and microRNA pathways, which can regulate PD-L1/PD-1 expression, leading to suppression of the adaptive immune response and allowing progression of the disease [43]. The expression of PD-L1 is increased in CIN1 and CINII-III as well as in HR-HPV-positive CeCa [19,25,44]. In fact, the proportions of CeCa tumors that express PD-L1 and PD-1 are 34.4–96% [25,43] and 46.97–60.82% [19,20], respectively, suggesting that the treatment of CeCa with checkpoint inhibitors may be a promising approach. However, understanding the mechanisms that promote PD-L1 expression in the TME is highly important because this pathway has been successfully targeted in cancer therapy. Hence, the current drug development strategies aim to overcome the failures of many drugs that are designed to block the PD-1/PD-L1 pathway and to address relapses that can occur in cancer patients after initial tumor regression [44,45].

PD-L1 can be expressed by various types of cells that reside in the TME, including tumor cells, immune cells, endothelial cells, and even immunosuppressor cells. Therefore, the expression of this ligand in tumors has been proposed to be a biomarker that can be used to predict the therapeutic effects of anti-PD-1 and anti-PD-L1 drugs [46]. It has been proposed that tumor cells express PD-L1 to mediate innate or adaptive resistance to immunity [47]. Innate resistance refers to the constitutive expression of PD-L1 in tumor cells due to amplification of the PDL1 gene or aberrant activation of oncogenic signaling pathways [48,49,50,51,52]; adaptive resistance to immunity refers to the expression of PD-L1 in tumors or immune cells in response to inflammatory factors that are secreted in the TME during antitumor immune responses, and IFN-γ is one of the main cytokines that is responsible for inducing the adaptive expression of PD-L1 [53]. However, several additional TME-resident cytokines that can upregulate PD-L1 expression, including IL-1a, IL-10, IL-27, and IL-32g [12,13,54,55], have been identified. Similarly, there is evidence showing that TGF-β regulates PD-L1/PD-1 expression in different tumor models [15,16,42] and that this cytokine promotes the enrichment of PD-L1 in exosomes derived from breast cancer tumor cells [56].

In previous studies, we found that CeCa cells infected with HPV-16 or HPV-18 produce greater amounts of Ado than do CeCa cells without HPV infection [37]. We also found that Ado induces the production of TGF-β1 in CeCa cells by interacting with A_2A_R/A_2B_R [35]. The present study provides the first lines of evidence showing that TGF-β1 production by CaSki and HeLa cells is induced by the interaction of Ado with A_2A_R and that A_2B_R acts in an autocrine manner to increase the expression of PD-L1 on the surface of these cells, promoting their immunosuppressive effects on CD8+ T lymphocytes. In addition, this factor promoted the expression of PD-1, strongly decreased the production of IFN-γ, and increased the production of TGF-β and IL-10 in activated CD8+ T lymphocytes (Figure 7).

Under the culture conditions used in this study, we observed that the increased expression of PD-L1 in CeCa cells was dependent on the presence of TGF-β in the supernatants. In fact, the increased TGF-β1 production that was observed after 24 h of culture corresponded with increased PD-L1 expression in CaSki and HeLa cells, and the significantly decreased PD-L1 expression that was observed after 72 h in both cell lines was associated with decreased TGF-β1 levels in the supernatants. This effect was also related to the increase in the expression of TGF-β1RII in CaSki and HeLa cells that were treated with Ado, which was doubled at 72 h. In addition, further analysis of whether adenosine-mediated signaling also has a feedback effect on the expression of A_2A_R and A_2B_R would be interesting, because Ado promotes angiogenesis, proliferation, migration, and metastasis of tumor cells via A_2A_R and A_2B_R [57].

In contrast, CaSki-Ado and HeLa-Ado supernatants also promoted the expression of PD-1, strongly decreased the production of IFN-γ, and increased the production of TGF-β and IL-10 in ATLs. However, although decreased expression of PD-1 or increased production of TGF-β and IL-10 in ATLs were largely reversed by the addition of the anti-TGF-β neutralizing antibody to the cultures, the decrease in IFN-γ production was only marginally reversed by the addition of this antibody, suggesting that other factors produced by Ado-treated CeCa cells play a strong role in the inhibition of T-cell activation by reducing the production of IFN-γ, consequently affecting the antitumor immune response. For example, in a recent study, CeCa patients with tumors expressing high PD-L1 or PD-L2 mRNA levels and exhibiting high IFN-γ signaling activity exhibited better overall survival than did CeCa patients with tumors expressing high PD-L1 expression and low IFN-γ activity [58], suggesting that the presence of factors that regulate IFN-γ production in the TME may play individual or synergistic roles in regulating PD-1 and PD-L1/PD-L2 expression [59].

The development of CeCa is strongly associated with the production of TGF-β1, whose expression is directly correlated with the degree of CeCa progression and antitumor immune response suppression [42]. TGF-β1 has been detected in the sera and tissues of patients infected with HPV-AR and in patients with CIN and CeCa [14,15,16,60]. TGF-β1 is an immunoregulatory factor involved in tumor progression and immunosuppression, and the sources of this factor in the TME include several types of stromal cells in addition to cancer cells [61,62]. We recently found that the secretion of TGF-β by Ado-treated CeCa cells maintains the expression of CD73 [35]. In this work, we observed that TGF-β produced by Ado-treated CeCa cells also induces the expression of PD-L1 in these cells, increasing the immunosuppressive effects of CeCa cells on CD8+ T cells. These findings suggest the existence of a feedback loop between adenosinergic activity and PD-L1 expression in cancer. In this feedback loop, the production of Ado and the signaling of this nucleoside in cancer cells can promote the expression of PD-L1 through TGF-β production, consequently favoring immune evasion and tumor progression in CeCa. In this regard, the correlation between PD-L1 and CD73 has been related to cancer prognosis [63,64], and blocking CD73 and PD-L1 has been shown to promote the T-cell response [65,66]; hence, these results offer a promising therapeutic strategy for cancer. In addition, through the design of bispecific recognition proteins, the inhibition of both TGF-β and PD-L1 was recently proposed for the treatment of cancer [67]. In this context, the use of bintrafusp alfa (BA), which is a protein that inhibits TGF-β and PD-L1, has been shown to decrease the size of tumors in a model of colorectal cancer [68]. Furthermore, encouraging results have been obtained in phase I clinical trials investigating the administration of this drug to patients with advanced CeCa [69]. Therefore, blocking the activity of the adenosinergic pathway to inhibit the production of Ado, as well as its signaling to induce the production of TGF-β and the subsequent expression of PD-L1 in the TME, could contribute to improving the design of combinatorial therapies against CeCa and other types of cancer. However, more information about the relationship between PD-L1 and TGF-β in CeCa is needed, such as, the analysis of their expression levels during CeCa progression to be considered as possible biomarkers.

## 4. Materials and Methods

### 4.1. CeCa Cell Lines

The CaSki (HPV-16+) and HeLa (HPV-18+) epithelial CeCa cell lines derived from human epidermoid carcinoma and adenocarcinoma, respectively, were obtained from the American Type Culture Collection (ATCC). These cell lines were free of mycoplasma, and their authenticity was verified by short tandem repeat (STR) genetic profiling. In previous studies, we found that these cell lines express CD73 on their surface and are capable of inhibiting the cytotoxic function of CD8+ T cells through the production of high amounts of Ado [37] and via the production of TGF-β [35].

### 4.2. Cell Culture

To determine the effect of Ado on the induction of PD-L1 expression in CeCa cells, 1 × 10^5^ CeCa cells were cultured in the presence of Ado (0, 0.1 mM, or 1 mM) (Sigma Aldrich, St. Louis, MO, USA) for 24 h, 48 h, 72 h, and 96 h. The expression of PD-L1 on CeCa cell membranes at each culture time point was analyzed using an APC-labeled anti-PD-L1 monoclonal antibody (clone #130021, R&D Systems, Minneapolis, MN, USA). To determine the role of the interaction of Ado with A_2_Rs in the induction of PD-L1 expression, selective antagonists of A_2B_R and A_2A_R, MRS1754 (Sigma-Aldrich, St. Louis, MO, USA) and ZM241385 (Sigma-Aldrich, St. Louis, MO, USA), or the selective A2 adenosine receptor agonist CGS21680 [38] were added at a concentration of 10 μM 30 min before Ado was added to the CeCa cell cultures, as previously reported [35,36]. The cultures were maintained in Opti-MEM culture media (OCM) composed of medium (Gibco, Oakland, CA, USA) supplemented with 1% bovine fetal serum (SFB; Gibco) dialyzed with a 12-kDa cutoff membrane (Sigma-Aldrich), 100 IU/mL penicillin, and 100 μg/mL streptomycin (Gibco) at 37 °C and 5% CO_2_.

The levels of TGF-β1 in the supernatants of CeCa cell cultures were determined using a Quantikine ELISA kit for human TGF-β1 (R&D Systems, Inc., Minneapolis, MN, USA) according to the manufacturer’s protocol. The role of CeCa cell-derived TGF-β in the autocrine induction of PD-L1 expression was analyzed after the addition of SB505124 (Sigma-Aldrich, USA), which is a selective inhibitor of the TGF-β1 receptor, at a final concentration of 25 µM. As a positive control for the induction of PD-L1 expression, CeCa cells were cultured for 24, 48, and 72 h in the presence of 20 ng/mL recombinant human TGF-β (rHuTGF-β, PeproTech, Rocky Hill, NJ, USA). To analyze the expression of PD-L1 in CeCa cells that were subjected to different treatments, 25,000 events were acquired via a Cytek Aurora Spectral flow cytometer (Cytek Biosciences, Fremont, CA, USA) and analyzed with the FlowJo V10 program (BD Biosciences, San Jose, CA, USA) after cellular debris was removed.

### 4.3. Isolation of Activated CD8+ T Lymphocytes (ATLs)

CD8+ T lymphocytes were harvested from normal donor peripheral blood mononuclear cells (PBMCs) by negative selection (EasySep Enrichment Cocktail, Stem Cell Technologies, Vancouver, BC, Canada). CD8+ T lymphocytes were cultured in supplemented ISCOVE’S medium (SIM), which was composed of ISCOVE’S-modified medium (Sigma–Aldrich, USA), 4 mM L-glutamine, 1 mM sodium pyruvate, 20 μM 2-mercaptoethanol, a mixture of nonessential amino acids (Gibco BRL, USA), antibiotics (100 U/mL penicillin and 100 μg/mL streptomycin), and 10% fetal bovine serum (SFB). Subsequently, the CD8+ T lymphocytes were activated by incubation with beads coated with anti-CD2/CD3/CD28 antibodies (Miltenyi Biotec GmbH, Bergisch Gladbach, Germany) at a bead:T-lymphocyte ratio of 2:1 in the presence of 100 U/mL recombinant interleukin-2 (hrIL-2) in a total volume of 200 μL, as previously reported [39].

### 4.4. Inhibition of ATLs by CeCa Cells

To determine the inhibitory effect of CeCa cells on CD8+ T-lymphocyte proliferation, CeCa cells cultured in the presence or absence of 1 mM Ado for 72 h were fixed with 4% paraformaldehyde (Sigma-Aldrich, USA), washed twice with SIM, and cocultured for 96 h with ATLs at a ratio of 1:5. ATL proliferation was assessed using the CellTiter 96^®^ Aqueous One Solution reagent (Promega, Madison, WI, USA) following the manufacturer’s instructions. The bioreduction of tetrazolium to formazan was analyzed at a wavelength of 490 nm with a GloMax plate reader (Promega, USA). The ATL proliferation index was determined every 24 h and calculated based on the optical density (OD) of the bioreduction of MTS to formazan induced by the ATLs in culture. The OD value that was obtained at the beginning of the culture was normalized to 1. In some cultures, a neutralizing anti-PD-L1 antibody (Novus, Chesterfield, MO, USA) was added according to the Novus protocol to inhibit the effect of PD-L1 on the proliferation of ATLs.

### 4.5. Cytotoxicity Assays

To analyze the ability of CeCa cells to inhibit the effector function of CTLs through PD-L1 expression, CTLs with specificity for the KLPQLCTEL antigenic peptide (residues 18–26) derived from the E6 protein of HPV-16 or TLHEYMLDL (residues 7–15) and YMLDLQPETT (residues 11–20) derived from the E7 protein of HPV-16, which bind specifically to the HLA-A0201 allele, expressed in CaSki cells [40], were previously generated using a previously reported method [41]. CaSki cells cultured for 72 h in the presence or absence of 1 mM Ado were fixed with 4% paraformaldehyde, washed twice with SIM, and cocultured for 48 h with CTLs at a ratio of 1:5. To block the inhibitory effect of PD-L1 on the cytotoxic activity of CTLs, in some cultures, a neutralizing anti-PD-L1 antibody (Novus, USA) was added according to the Novus protocol. CTLs cultured in the absence of CaSki cells were used as a control (Ctl). CaSki cells were marked with 5(6)-carboxyfluorescein diacetate N-succinimidyl ester (CFSE) (Sigma-Aldrich, St. Louis, MO, USA), and CTLs were labeled with a CD8-APC antibody (R&D Systems, Inc., Minneapolis, MN, USA) following the protocol provided by the supplier. The cell viability solution 7-AAD (Sigma-Aldrich, St. Louis, MO, USA) was used to measure cell lysis. Effector and target cells were incubated for 4 h at a ratio of 2:1. After the removal of cellular debris, viable cells were analyzed using a Cytek Aurora Spectral flow cytometer (Cytek Biosciences, USA), and a minimum of 100,000 events were collected. In some cases, 5% hydrogen peroxide was used for the total lysis of target cells. The percentage of lysis was calculated according to the following formula: % cytotoxicity = 100 × [experimental lysis (CFSE+, 7-AAD+) − basal lysis (CFSE+, 7AAD+)].

### 4.6. Assessment of PD-1, TGF-β1 and IL-10 Expression in ATLs

A total of 2.5 × 10^5^ ATLs per well were cultured in a 24-well plate (Corning, NY, USA) for 48 h in SIM supplemented with 30% supernatant from CeCa cells (CeCa-Sup) or CeCa cells that were previously stimulated with 1 mM Ado (CeCa/Ado-Sup) for 72 h. To neutralize the effect of TGF-β in the supernatants of CeCa cells that were treated with Ado, 1 μg/mL rabbit anti-human TGF-β1, TGF-β2, and TGF-β3 neutralizing antibodies (anti-TGF-β; R&D Systems, Minneapolis, MN, USA) were added. The expression levels of PD-1, IFN-γ, TGF-β1, and IL-10 in ATLs cultured in the absence of CeCa-Sup or CeCa/Ado-Sup were considered the control (Ctl) levels. During the last 4 h of culture, brefeldin-A (Sigma-Aldrich, St. Louis, MO, USA) was added at a final concentration of 10 μM to measure the intracellular levels of IFN-γ, TGF-β1, and IL-10. Subsequently, the cells were collected, fixed, and permeabilized using a Cytofix/Cytoperm Kit (BD Biosciences, San Jose, CA, USA). T cells were labeled with anti-IFN-γ/FITC and anti-CD8/APC (R&D Systems, Inc., Minneapolis, MN, USA), anti-human IL-10/FITC (Novus Biologicals, USA), and anti-human TGF-β/PE (LifeSpan, Australia) monoclonal antibodies. The cells were incubated for 30 min at 4 °C and washed three times with PBS. To determine the expression of PD-1, ATLs that were cultured under the above-described conditions in the absence of brefeldin-A were labeled with anti-CD8/Alexa Fluor 405 (R&D, USA) and anti-PD-1/PE (Novus, USA) monoclonal antibodies according to the manufacturer’s protocol. To determine the effect of TGF-β on the expression of PD-1, IFN-γ, TGF-β1, and IL-10, ATLs were cultured for 48 h in the presence of 20 ng/mL rHuTGF-β1. To analyze the cells, 25,000 events were acquired using a Cytek Aurora Spectral flow cytometer (Cytek Biosciences, USA) after removing cellular debris, and the data were analyzed with FlowJo 10.4 software.

### 4.7. Statistical Analysis

All numerical data are shown as the average values ± SEMs of three independent experiments. The statistical significance was calculated using the one-way ANOVA (nonparametric) test to compare the mean rank of problem data with the mean rank of control data using the GraphPad Prism software version 8 (GraphPad Software, San Diego, CA, USA). Differences were considered significant if *p* < 0.05.

## 5. Conclusions

The present study provides evidence showing that the interaction of adenosine with A_2A_R/A_2B_R increases the immunosuppressive capacity of CeCa cells via the production of TGF-β, which acts in an autocrine manner to induce the expression of PD-L1 on CeCa cells and acts in a paracrine manner to inhibit the activation of CD8+ lymphocytes through the induction of PD-1 expression; these phenomena result in decreased production of IFN-γ and increased production of TGF-β and IL-10. The findings of this study suggest that PD-L1 and TGF-β may be clinically important biomarkers and serve as therapeutic targets.

## Figures and Tables

**Figure 1 pharmaceuticals-17-00397-f001:**
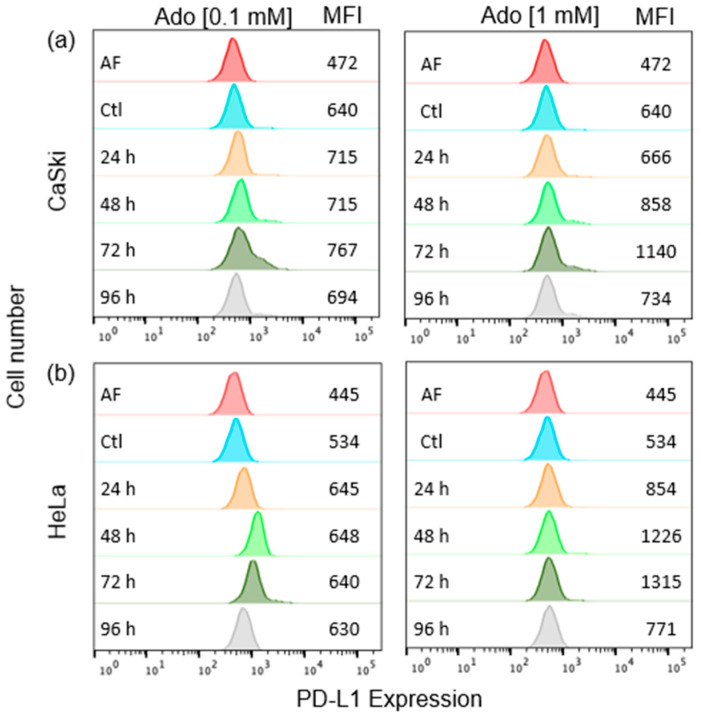
Effect of Ado on the expression of PD-L1 in CeCa cells. CaSki (**a**) and HeLa (**b**) cells (1 × 10^5^ of each) were cultured in the presence of 0.1 mM or 1 mM Ado for 96 h. The baseline expression level of PD-L1 in CeCa cells was considered the control (Ctl) level. PD-L1 expression was analyzed via flow cytometry every 24 h. Representative data from three independent experiments are shown. MFI, mean fluorescence intensity.

**Figure 2 pharmaceuticals-17-00397-f002:**
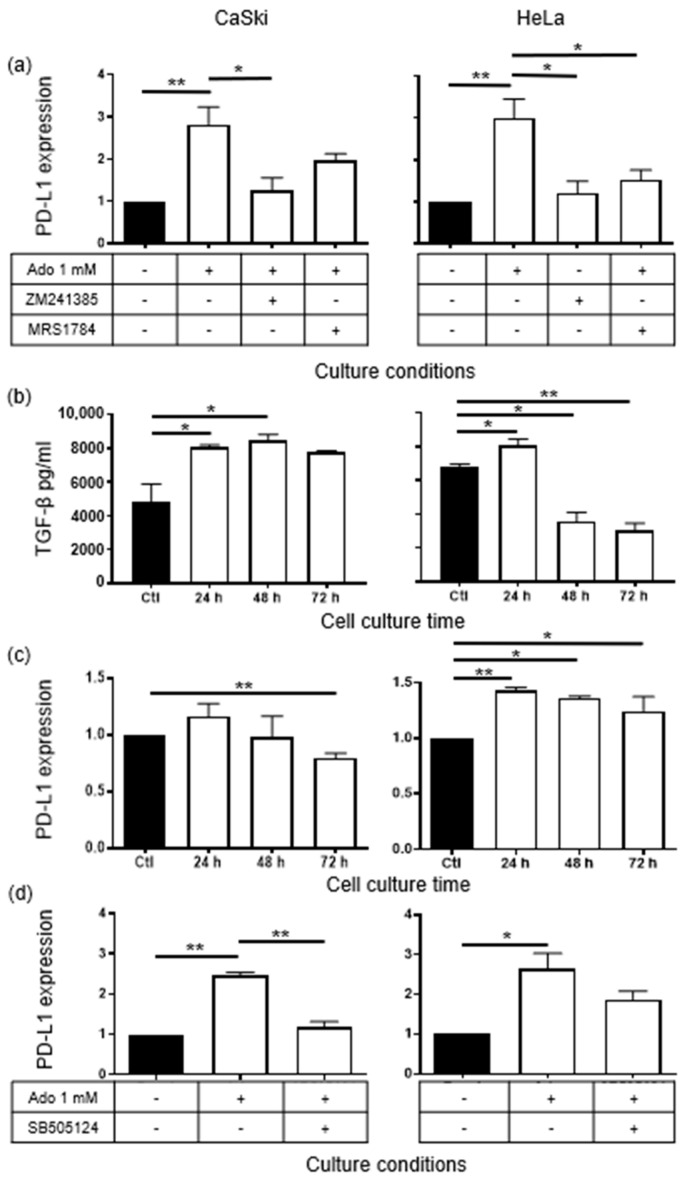
Ado increases PD-L1 expression on CeCa cells by interacting with A_2A_R and A_2B_R to induce the production of TGF-β1. (**a**) CaSki and HeLa cells (1 × 10^5^ each) were first incubated for 30 min in the presence of 10 µM ZM241385 and MRS1754, which are selective antagonists for A_2A_R and A_2B_R, respectively. The cells were then cultured for 72 h in the presence or absence of 1 mM Ado. Cells that were cultured in the absence of both Ado and AR antagonists were used as controls (black bars). (**b**) The levels of TGF-β in the supernatants of CeCa cells cultured with 1 mM Ado for 24 h, 48 h and 72 h were determined via ELISA. The TGF-β levels in the supernatants of CeCa cells that were cultured in the absence of Ado were considered control levels (black bars). (**c**) CeCa cells were cultured in the presence of 20 ng of rHuTGF-β1. PDL-1 expression in CeCa cells was determined at 24 h, 48 h, and 72 h. CeCa cells that were cultured in the absence of rHuTGF-β1 were used as a control (black bars). (**d**) CeCa cells were cultured for 72 h in the presence or absence of 1 mM Ado and in the presence or absence (black bars) of 25 μM SB-505124, which is a selective TGF-β1 receptor inhibitor. The statistical significance was calculated using the one-way ANOVA (nonparametric) test. Representative data from three independent experiments (means ± SEMs) are shown. * *p* < 0.05; ** *p* < 0.001. +, presence. -, absence.

**Figure 3 pharmaceuticals-17-00397-f003:**
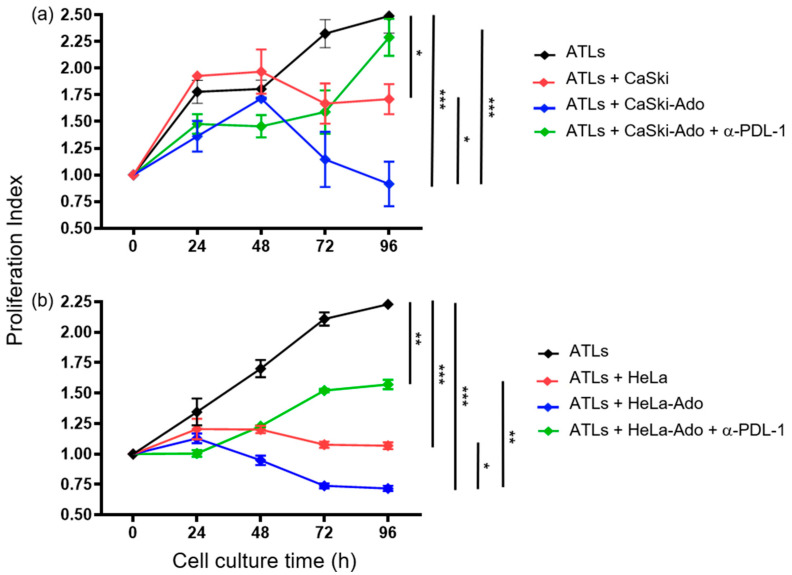
Ado-stimulated CeCa cells inhibit the proliferation of activated CD8+ T cells by expressing PD-L1. The proliferation indices (PIs) of activated CD8+ T lymphocytes (ATLs) that were cultured for 24, 48, 72, and 96 h in the presence (red lines) or absence (black lines) of CaSki (**a**) and HeLa (**b**) cells that were first cultured in either the presence (blue lines) or the absence (red lines) of 1 mM Ado are shown. In some cultures, a neutralizing anti-PD-L1 antibody was added to inhibit the effect of PD-L1 on the proliferation of ATLs cultured with CeCa cells that were previously incubated in the presence (green lines) of 1 mM Ado. The PIs of the ATLs at the beginning of the culture were set to 1. The statistical significance was calculated using the one-way ANOVA (nonparametric) test. Representative data from three independent experiments (means ± SEMs) are shown. * *p* < 0.05, ** *p* < 0.01, *** *p* < 0.001.

**Figure 4 pharmaceuticals-17-00397-f004:**
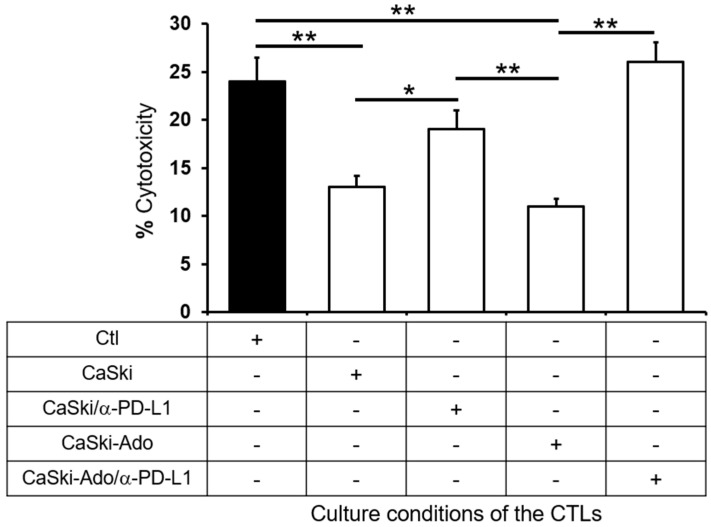
Ado-stimulated CeCa cells inhibit the effect of cytotoxic T lymphocytes by expressing PD-L1. CaSki cells or CaSki-Ado cells were fixed with paraformaldehyde and cocultured for 48 h at a ratio of 1:5 with CTLs (CD8+ T lymphocytes) specific for the KLPQLCTEL antigenic peptide (residues 18–26) derived from the E6 protein of HPV-16 or TLHEYMLDL (residues 7–15) and YMLDLQPETT (residues 11–20) derived from the E7 protein of HPV-16. To block the inhibitory effect of PD-L1 on the cytotoxic activity of CTLs, in some cultures, a neutralizing anti-PD-L1 antibody was added. CTLs cultured in the absence of CaSki cells were used as a control Ctl (Black bar). CTL activity was evaluated by measuring the viability of cells stained with CFSE and 7-AAD, as indicated in the Methods section. The statistical significance was calculated using the one-way ANOVA (nonparametric) test. The data represent three independent experiments; the means ± SEMs are shown. * *p* < 0.05; ** *p* < 0.001. +, presence. -, absence.

**Figure 5 pharmaceuticals-17-00397-f005:**
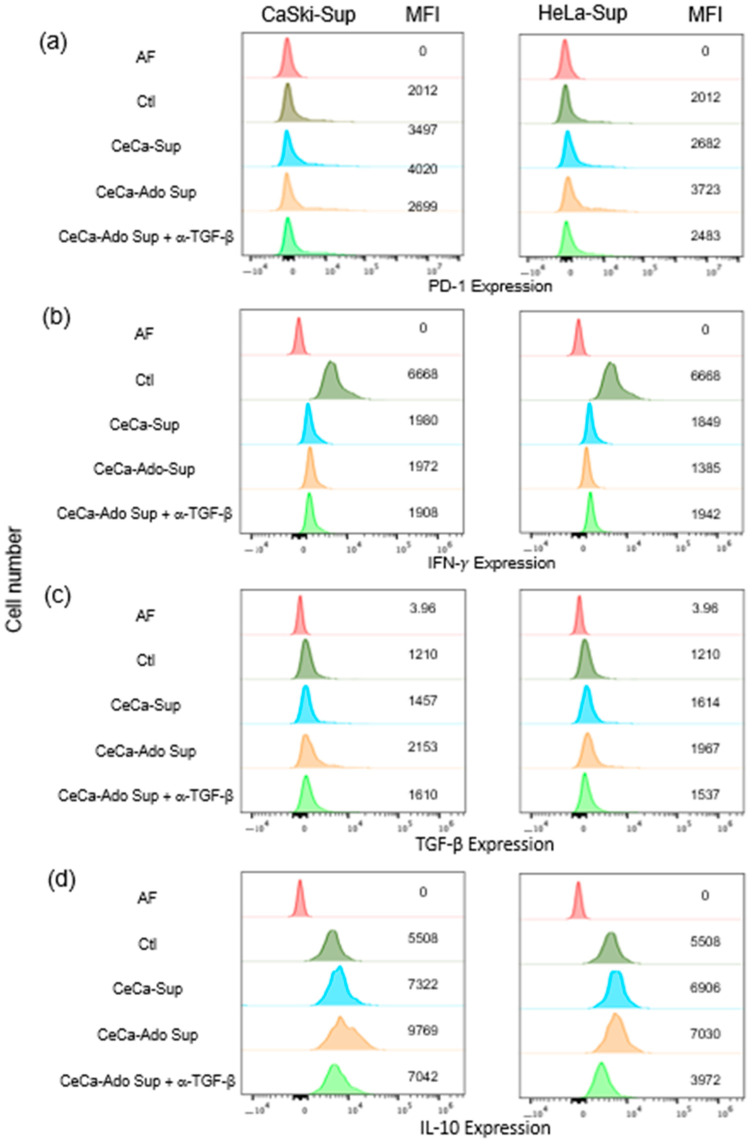
Effect of CeCa-Ado supernatants on the activation of CD8+ T cells. CD8+ T cells (2.5 × 10^5^) were cultured for 48 h in the presence of beads coated with anti-CD2/CD3/CD28 antibodies at a ratio of 2:1 with or without 30% supernatant (Sup) from CeCa cells or CeCa-Ado cells and with or without 1 μg/mL anti-TGF-β1 neutralizing antibody. The expression of PD-1 (**a**), IFN-γ (**b**), TGF-β (**c**), and IL-10 (**d**) in ATLs was analyzed as described in the Materials and Methods section. ATLs that were cultured without CeCa or CeCa-Ado Sup were considered controls (Ctls). Representative data from three independent experiments are shown. MFI, mean fluorescence intensity.

**Figure 6 pharmaceuticals-17-00397-f006:**
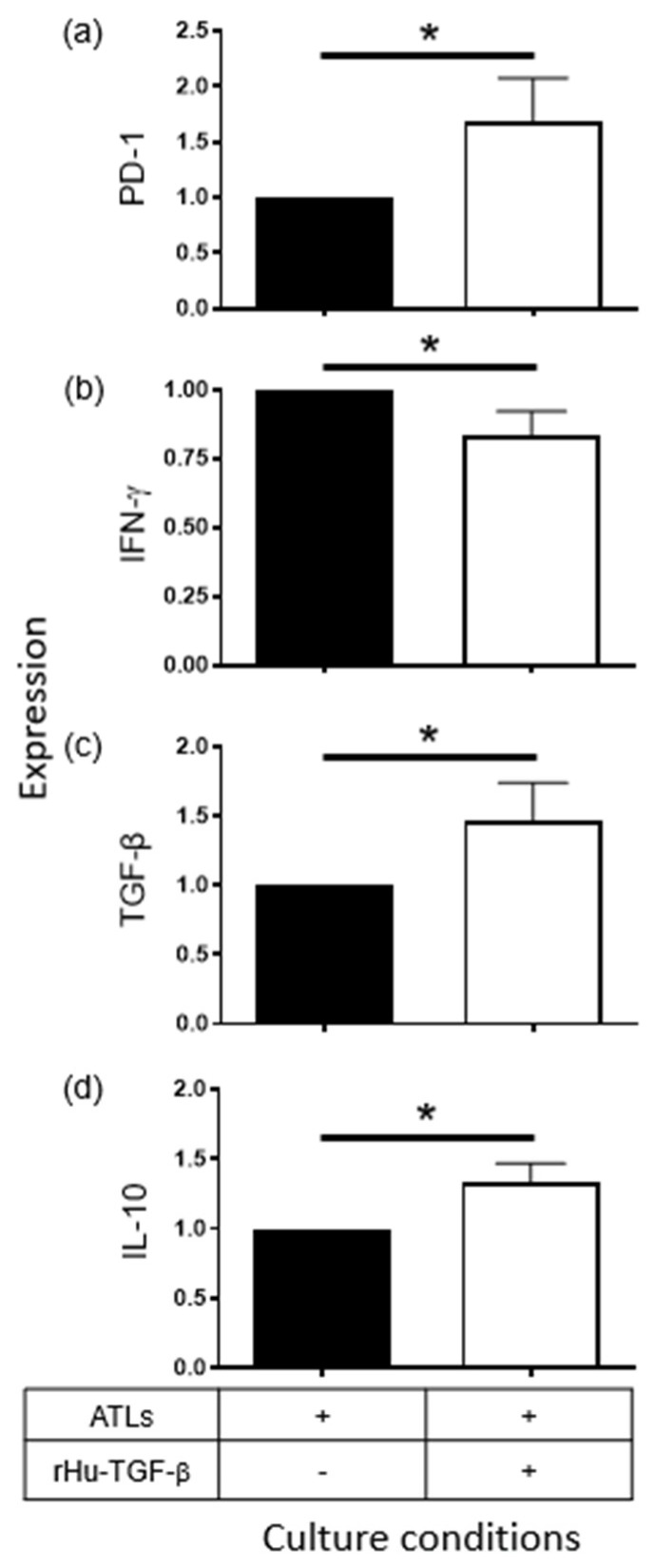
Effect of TGF-β on the activation of CD8+ T cells. CD8+ T cells (2.5 × 10^5^) were cultured for 48 h in the presence of beads coated with anti-CD2/CD3/CD28 antibodies at a ratio of 2:1 in the presence or absence of 20 ng/mL rHuTGF-β. The expression of PD-1 (**a**), IFN-γ (**b**), TGF-β (**c**) and IL-10 (**d**) in ATLs was analyzed as described in the Materials and Methods section. The expression levels of PD-1, IFN-γ, TGF-β, and IL-10 in ATLs that were cultured without rHuTGF-β were considered control levels (black bars) and were set to 1. Representative data from three independent experiments (means ± SEMs) are shown. The statistical significance was calculated using the one-way ANOVA (nonparametric) test. * *p* < 0.05 compared to the control.

**Figure 7 pharmaceuticals-17-00397-f007:**
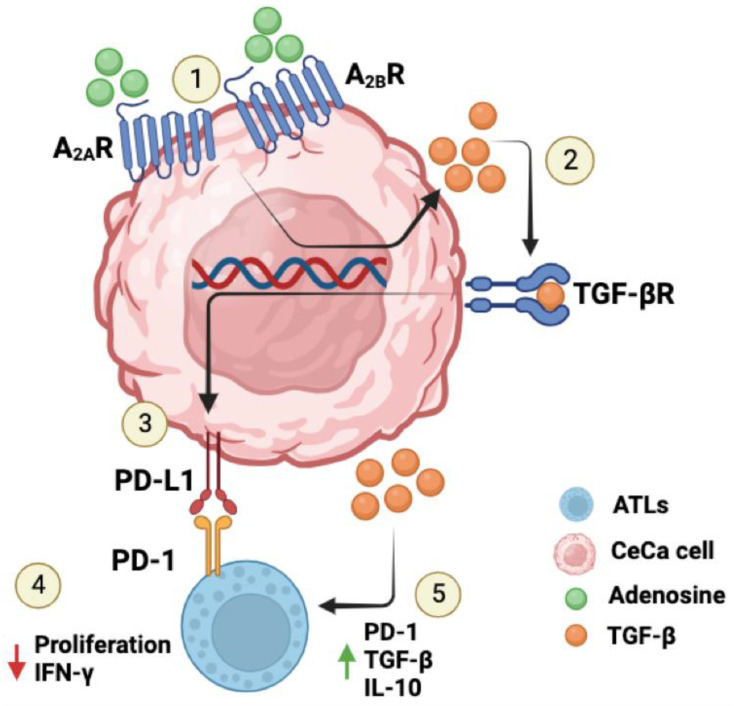
Proposed mechanism by which Ado increases the immunosuppressive effects of CeCa cells by increasing PD-L1 expression and TGF-β production. The interaction of Ado with A_2A_R and A_2B_R (1) induces the production of TGF-β1 by CeCa cells (2). This cytokine acts in an autocrine manner, increasing the expression of PD-L1 on the membrane (3); as a result, the immunosuppressive effect of CeCa cells on activated CD8+ T lymphocytes (ATLs) is increased. Furthermore, TGF-β inhibits ATLs by decreasing the production of IFN-γ (4) and increasing the expression of PD-1 and the production of TGF-β and IL-10 (5).

## Data Availability

Data are available upon reasonable request to the corresponding author.

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
