# Peer review of "Adenosine Increases the Immunosuppressive Capacity of Cervical Cancer Cells by Increasing PD-L1 Expression and TGF-β Production through Its Interaction with A2AR/A2BR"

_pharmaceuticals, 2024, doi:10.3390/ph17030397_

Round 1

Reviewer 1 Report

Comments and Suggestions for Authors

In their study, the authors elucidated that adenosine enhances the immunosuppressive capacity of cervical cancer (CeCa) cells through the upregulation of PD-L1 expression and the production of TGF-β, facilitated by its interaction with A2AR/A2BR. Additionally, it was demonstrated that supernatants from adenosine-treated CeCa cells inhibit the activation of CD8+ T lymphocytes and increase PD-1 expression. These findings suggest that the adenosine pathway plays a crucial role as a mechanism utilized by CeCa cells to evade immune responses.

The results of this study are based on rigorous experimental methodologies and statistical analysis. The authors have employed multiple experimental approaches to demonstrate that the immunosuppressive effect is dependent on the increase in PD-L1 expression and TGF-β production mediated by adenosine. The interpretation of the results is logical and clearly articulated. However, the addition of some further experiments and supplementary information would strengthen the support for these findings. Considering the significance and quality of this study, I recommend a minor revision before publication. Below, some constructive comments and suggested revisions necessary for publication are listed.

Suggested Revisions:

#1: Further clarification of experimental design: It is suggested that detailed descriptions of the characteristics and reasons for selecting the used cell lines be added.

#2: Addition of information on statistical analysis: It is recommended to reference and explain the specific statistical methods used to determine the statistical significance of each experiment within the text.

This research represents an important step towards deepening the understanding of immunotherapy for cervical cancer. Implementing the above revisions will likely enhance the reliability and impact of the study findings.

Comments on the Quality of English Language

In the manuscript, there are several instances of long sentences packed with substantial information. It is suggested that splitting these into shorter sentences could facilitate easier comprehension and clarify the main points for the reader. Simplifying complex information into more digestible pieces can significantly enhance the overall readability and accessibility of the text, potentially increasing its impact on the intended audience.

Author Response

Reviewer 1

Comments and Suggestions for Authors

In their study, the authors elucidated that adenosine enhances the immunosuppressive capacity of cervical cancer (CeCa) cells through the upregulation of PD-L1 expression and the production of TGF-β, facilitated by its interaction with A2AR/A2BR. Additionally, it was demonstrated that supernatants from adenosine-treated CeCa cells inhibit the activation of CD8+ T lymphocytes and increase PD-1 expression. These findings suggest that the adenosine pathway plays a crucial role as a mechanism utilized by CeCa cells to evade immune responses.

The results of this study are based on rigorous experimental methodologies and statistical analysis. The authors have employed multiple experimental approaches to demonstrate that the immunosuppressive effect is dependent on the increase in PD-L1 expression and TGF-β production mediated by adenosine. The interpretation of the results is logical and clearly articulated. However, the addition of some further experiments and supplementary information would strengthen the support for these findings. Considering the significance and quality of this study, I recommend a minor revision before publication. Below, some constructive comments and suggested revisions necessary for publication are listed.

Suggested Revisions:

#1: Further clarification of experimental design: It is suggested that detailed descriptions of the characteristics and reasons for selecting the used cell lines be added.

Our answer: As recommended, detailed descriptions of the characteristics and reasons for selecting the cell lines used in this study were added to the text (Materials and Methods section, lines 434-440).

#2: Addition of information on statistical analysis: It is recommended to reference and explain the specific statistical methods used to determine the statistical significance of each experiment within the text.

Our answer: As recommended, information on the statistical methods used for analysis of the data from each experiment was added to the Materials and Methods section (lines 548-554) and in the figure legends.

This research represents an important step toward deepening the understanding of immunotherapy for cervical cancer. Implementing the above revisions will likely enhance the reliability and impact of the study findings.

Comments on the Quality of English Language

In the manuscript, there are several instances of long sentences packed with substantial information. It is suggested that splitting these into shorter sentences could facilitate easier comprehension and clarify the main points for the reader. Simplifying complex information into more digestible pieces can significantly enhance the overall readability and accessibility of the text, potentially increasing its impact on the intended audience.

Our answer: As recommended, long sentences were divided into shorter sentences to facilitate easier comprehension and clarity of the ideas. The information was also simplified. The manuscript of this new version was reviewed and corrected by American Journal Experts, the Editing Certificate is attached.

Reviewer 2 Report

Comments and Suggestions for Authors

In this article authors investigated the adenosine axis and its relationship with the immunosuppressive environment via PD-L1 expression and also the interaction of TGF-beta through its interaction with A2aR/A2BR axis. The manuscript is well written but can be improved further. Following are some comments. 

1. In figure 1 author should change the x-axis to log scale for a better view. 

2. Authors did not presented any cytotoxic data. Their investigation is also limited to CD8T cells.

3. To better elaborate the scope of their work they can also include some specific CAR mediated killing or serial killing with or without inhibitors and their antagonist. 

4. The half life of adenosine is shorter, did authors have an understand of how long this adenosine was active after addition. Their are better analogues available to use to study adenosine did authors consider those avenues. 

Comments on the Quality of English Language

The english language is OK 

Author Response

Reviewer 2

In this article authors investigated the adenosine axis and its relationship with the immunosuppressive environment via PD-L1 expression and also the interaction of TGF-beta through its interaction with A2aR/A2BR axis. The manuscript is well written but can be improved further. Following are some comments.

  1. In figure 1 author should change the x-axis to log scale for a better view.

Our answer: As recommended, the x-axis in Figure 1 was changed to a log scale.

  1. Authors did not presented any cytotoxic data. Their investigation is also limited to CD8T cells.

Our answer: The immunosuppressive effect of PD-L1 in Ado-stimulated CeCa cells and on the proliferation of activated T lymphocytes was also assayed using cytotoxic T lymphocytes stimulated with antigenic peptides derived from the E6 and E7 proteins of HPV-16, which specifically bind to the HLA-A2 allele expressed on CaSki (HPV-16+) cells (Lines 217-227, Figure 4).

  1. To better elaborate the scope of their work they can also include some specific CAR mediated killing or serial killing with or without inhibitors and their antagonist.

Our answer: We do not have CAR-T cells; however, as mentioned above, the immunosuppressive effect of Ado-stimulated CeCa cells via PD-L1 was also assayed using cytotoxic T lymphocytes stimulated with antigenic peptides derived from the E6 and E7 proteins of HPV-16, which specifically bind to the HLA-A2 allele expressed on CaSki (HPV-16+) cells. The effect of PD-L1 observed in these assays was corroborated in the presence of the anti-PD-L1 monoclonal antibody (Lines 217-227, Figure 4).

  1. The half life of adenosine is shorter, did authors have an understand of how long this adenosine was active after addition. Their are better analogues available to use to study adenosine did authors consider those avenues.

Our answer: We agree with your comment. For this reason, we analyzed the effect of two concentrations of Ado (0.1 mM and 1 mM) on the expression of PD-L1 in CeCa cells cultured for 24 h, 48 h, 72 h and 96 h. As shown in Figure 1a and 1b, Ado increased the expression of PD-L1 in both cell lines, and this effect depended on the duration of culture with Ado and the concentration of this nucleoside. Based on these results, we selected 1 mM Ado for 72 h for the subsequent experiments.

In contrast, considering that the Ado signaling-induced expression of PD-L1 on CeCa cells was dependent on A2AR or A2BR, we also assayed the effect of CGS21680, a selective A2 agonist adenosine receptor, on the expression of PD-L1 in CeCa cells. As shown in Figure S2, CGS21680 induced a significant increase in PD-L1 expression in CaSki and HeLa cells, comparable to that observed in CeCa cells cultured in the presence of 1 mM Ado.

Comments on the Quality of English Language

The english language is OK

Reviewer 3 Report

Comments and Suggestions for Authors

The authors presented data on immunosuppressive capacity of cervical cancer cells by increasing PD-L1 expression and the production of TGF-b through its interaction with A2AR/A2BR which are interesting from the point of view of the use of such mechanisms in the treatment of cancer through immunomodulation.

Some previous work should be cited in the Introduction section such as Molecular cell vol. 76,3 (2019): 359-370. doi:10.1016/j.molcel.2019.09.030, Journal for immunotherapy of cancer vol. 6,1 57. 18 Jun. 2018, doi:10.1186/s40425-018-0360-8

In the presented graphs obtained using flow cytometry, it would be informative to include the distribution of events on the plot. Or include this data in supplementary materials. 

Conclusion must be extended to demonstrate all the findings in details.

Author Response

Reviewer 3

The authors presented data on immunosuppressive capacity of cervical cancer cells by increasing PD-L1 expression and the production of TGF-b through its interaction with A2AR/A2BR which are interesting from the point of view of the use of such mechanisms in the treatment of cancer through immunomodulation.

Some previous work should be cited in the Introduction section such as Molecular cell vol. 76,3 (2019): 359-370. doi:10.1016/j.molcel.2019.09.030, Journal for immunotherapy of cancer vol. 6,1 57. 18 Jun. 2018, doi:10.1186/s40425-018-0360-8

Our answer: As recommended, relevant information obtained from doi:10.1016/j.molcel.2019.09.030 (lines 70-73, reference [11]) and doi:10.1186/s40425-018-0360-8 (lines 99-103, reference [34]) was cited in the text.

In the presented graphs obtained using flow cytometry, it would be informative to include the distribution of events on the plot. Or include this data in supplementary materials.

Our answer: As recommended, data about the distribution of events on the plot related to the effect of Ado on PD-L1 expression in CeCa cells through A2AR/A2BR were included in the supplementary materials (Figure S1).

Conclusion must be extended to demonstrate all the findings in details.

Our answer: As recommended, the extended conclusion describes all the findings in detail (lines 555-562).

Reviewer 4 Report

Comments and Suggestions for Authors

The study presented in this manuscript investigates the role of adenosine in modulating the immunosuppressive capacity of cervical cancer cells. The authors provide evidence that adenosine increases the expression of PD-L1 in cervical cancer cells by interacting with A2AR/A2BR receptors and stimulating the production of TGF-β1. The findings suggest a potential mechanism by which cervical cancer cells evade the immune system and propose the adenosinergic pathway as a therapeutic target. Overall, the study is well-conducted and the results are interesting. However, there are some concerns and questions that need to be addressed before a final decision can be made.

The introduction provides a good background on cervical cancer and its association with HPV infection. However, it would be helpful to include more information on the current understanding of the role of PD-L1 and TGF-β in cervical cancer immunosuppression. Additionally, please provide a rationale for investigating the adenosinergic pathway in this context.

The methods section lacks sufficient details regarding the experimental procedures and materials used. Please provide more information on the cell culture conditions, treatment protocols, and specific reagents used for the experiments. This will help readers better understand the experimental setup and replicate the study if needed.

The results section presents clear data on the upregulation of PD-L1 expression in cervical cancer cells treated with adenosine. However, the functional implications of this upregulation are not well addressed. Please discuss the potential effects of increased PD-L1 expression on the immune response and the significance of these findings in the context of cervical cancer progression.

The discussion section provides a good interpretation of the results and their implications. However, it would be beneficial to include a more thorough analysis of the limitations and potential future directions of the study. Additionally, please address the clinical relevance of targeting the adenosinergic pathway as a therapeutic strategy for cervical cancer.

The manuscript lacks statistical analysis for the presented data. It is essential to include statistical tests to support the significance of the findings. Please provide statistical analysis and include p-values or confidence intervals where appropriate.

Please proofread the manuscript for grammatical errors and typos. There are several instances of incomplete sentences and missing punctuation that need to be corrected.

The conclusion section should be revised to summarize the key findings and their implications more clearly. Additionally, please address the major limitations of the study.

Questions:

Did the authors investigate the expression levels of A2AR and A2BR receptors in cervical cancer cells? If so, please provide the results and discuss the potential implications of receptor expression on the response to adenosine.

What is the rationale for using ZM241385 and MRS1754 as specific antagonists for A2AR and A2BR receptors, respectively? Please provide information on the selectivity and potency of these antagonists.

The authors mention the use of an anti-TGF-β neutralizing antibody to reverse the effect of CeCa-Ado Sup on PD-1 expression. Could the authors provide more details on the experimental setup and the results obtained with this antibody?

the manuscript requires major revisions before it can be considered for publication. The concerns regarding the lack of detailed methods, statistical analysis, and the need for further discussion on the clinical relevance and limitations of the study must be addressed. Additionally, the manuscript should be carefully proofread to correct grammatical errors and improve readability. Once the revisions have been made, the manuscript can be re-evaluated for final consideration.

Comments on the Quality of English Language

the manuscript should be carefully proofread to correct grammatical errors and improve readability.

Author Response

Reviewer 4

The study presented in this manuscript investigates the role of adenosine in modulating the immunosuppressive capacity of cervical cancer cells. The authors provide evidence that adenosine increases the expression of PD-L1 in cervical cancer cells by interacting with A2AR/A2BR receptors and stimulating the production of TGF-β1. The findings suggest a potential mechanism by which cervical cancer cells evade the immune system and propose the adenosinergic pathway as a therapeutic target. Overall, the study is well-conducted and the results are interesting. However, there are some concerns and questions that need to be addressed before a final decision can be made.

The introduction provides a good background on cervical cancer and its association with HPV infection. However, it would be helpful to include more information on the current understanding of the role of PD-L1 and TGF-β in cervical cancer immunosuppression. Additionally, please provide a rationale for investigating the adenosinergic pathway in this context.

Our answer: Information on the current understanding of the role of PD-L1 and TGF-β in CeCa immunosuppression was included in the Introduction section (lines 78-87), and the relationship between adenosinergic activity and CeCa was also included (lines 99-107).

The methods section lacks sufficient details regarding the experimental procedures and materials used. Please provide more information on the cell culture conditions, treatment protocols, and specific reagents used for the experiments. This will help readers better understand the experimental setup and replicate the study if needed.

Our answer: As recommended, more information on the cell culture conditions, treatment protocols, and specific reagents used for the experiments was included.

The results section presents clear data on the upregulation of PD-L1 expression in cervical cancer cells treated with adenosine. However, the functional implications of this upregulation are not well addressed. Please discuss the potential effects of increased PD-L1 expression on the immune response and the significance of these findings in the context of cervical cancer progression.

Our answer: As recommended, the potential effects of increased PD-L1 expression on the immune response and the significance of these findings in the context of CeCa progression were included in the Discussion section (lines 399-415).

The discussion section provides a good interpretation of the results and their implications. However, it would be beneficial to include a more thorough analysis of the limitations and potential future directions of the study. Additionally, please address the clinical relevance of targeting the adenosinergic pathway as a therapeutic strategy for cervical cancer.

Our answer: As recommended, analysis of the limitations and potential future directions of the study, as well as the clinical relevance of targeting the adenosinergic pathway as a therapeutic strategy for cervical cancer, was included in the Discussion section (lines 419-427).

The manuscript lacks statistical analysis for the presented data. It is essential to include statistical tests to support the significance of the findings. Please provide statistical analysis and include p-values or confidence intervals where appropriate.

Our answer: As recommended, information on the statistical methods used for analysis of the data from each experiment, including p values, was added to the Materials and Methods section (lines 548-554) and in the figure legends.

Please proofread the manuscript for grammatical errors and typos. There are several instances of incomplete sentences and missing punctuation that need to be corrected.

Our answer: As recommended, the grammatical typographical errors in manuscript were corrected. The manuscript of this new version was reviewed and corrected by American Journal Experts, the Editing Certificate is attached.

The conclusion section should be revised to summarize the key findings and their implications more clearly. Additionally, please address the major limitations of the study.

Our answer: As recommended, the extended conclusion includes all the findings in detail as well as the major limitations of the study (lines 555-562).

Questions:

Did the authors investigate the expression levels of A2AR and A2BR receptors in cervical cancer cells? If so, please provide the results and discuss the potential implications of receptor expression on the response to adenosine.

Our answer: We did not determine the expression levels of the A2AR and A2BR receptors in CeCa cells. However, in the discussion section (lines 380-383), we argue the importance of this effect due to the relevant role of adenosine signaling through these receptors in tumor progression.

What is the rationale for using ZM241385 and MRS1754 as specific antagonists for A2AR and A2BR receptors, respectively? Please provide information on the selectivity and potency of these antagonists.

Our answer: The affinity of the selective antagonist ZM241385 for the A2A receptor is high, with a value of Ki = 0.2 nM. However, the affinity of the selective antagonist MRS1754 for the A2B receptor is low, with a Ki value > 1000 nM. However, to block A2AR and A2BR on CeCa cells, we used a concentration of 10 mM of each antagonist.

The authors mention the use of an anti-TGF-β neutralizing antibody to reverse the effect of CeCa-Ado Sup on PD-1 expression. Could the authors provide more details on the experimental setup and the results obtained with this antibody?

Our answer: To neutralize the effect of TGF-β in the supernatants of CeCa cells that were treated with Ado, 1 μg/ml rabbit anti-human TGF-β1, TGF-β2, and TGF-β3 neutralizing antibodies (anti-TGF-β; R&D Systems, Minneapolis, MN, USA) were added (lines 527-530). The results obtained with this antibody are described in the Results section (lines 266-272).

the manuscript requires major revisions before it can be considered for publication. The concerns regarding the lack of detailed methods, statistical analysis, and the need for further discussion on the clinical relevance and limitations of the study must be addressed. Additionally, the manuscript should be carefully proofread to correct grammatical errors and improve readability. Once the revisions have been made, the manuscript can be re-evaluated for final consideration.

Comments on the Quality of English Language

the manuscript should be carefully proofread to correct grammatical errors and improve readability.

Round 2

Reviewer 2 Report

Comments and Suggestions for Authors

The authors have addressed all the concerns and I have no further comments.

Reviewer 4 Report

Comments and Suggestions for Authors

accepted